# Path Sample-Analytic Gradient Estimators for Stochastic Binary Networks

**Alexander Shekhovtsov**
Czech Technical University in Prague
shekhovt@cmp.felk.cvut.cz

**Viktor Yanush**
Lomonosov Moscow State University
yanushviktor@gmail.com

**Boris Flach**
Czech Technical University in Prague
flachbor@cmp.felk.cvut.cz

## Abstract

In neural networks with binary activations and or binary weights the training by gradient descent is complicated as the model has piecewise constant response. We consider stochastic binary networks, obtained by adding noises in front of activations. The expected model response becomes a smooth function of parameters, its gradient is well defined but it is challenging to estimate it accurately. We propose a new method for this estimation problem combining sampling and analytic approximation steps. The method has a significantly reduced variance at the price of a small bias which gives a very practical tradeoff in comparison with existing unbiased and biased estimators. We further show that one extra linearization step leads to a deep straight-through estimator previously known only as an ad-hoc heuristic. We experimentally show higher accuracy in gradient estimation and demonstrate a more stable and better performing training in deep convolutional models with both proposed methods.

## 1 Introduction

Neural Networks with binary weights and binary activations are very computationally efficient. Rastegari et al. [24] report up to $58\times$ speed-up compared to floating point computations. There is a further increase of hardware support for binary operations: matrix multiplication instructions in recent NVIDIA cards, specialized projects on spike-like (neuromorphic) computation [3, 7], *etc*.

Binarized (or more generally quantized) networks have been shown to close up in performance to real-valued baselines [2, 22, 25, 28, 12, 6]. We believe that good training methods can improve their performance further. The main difficulty with binary networks is that unit outputs are computed using sign activations, which renders common gradient descent methods inapplicable. Nevertheless, experimentally oriented works ever so often define the lacking gradients in these models in a heuristic way. We consider the more sound approach of stochastic Binary Networks (SBNs) [19, 23]. This approach introduces injected noises in front of all sign activations. The network output becomes smooth in the expectation and its derivative is well-defined. Furthermore, injecting noises in all layers makes the network a deep latent variable model with a very flexible predictive distribution.

Estimating gradients in SBNs is the main problem that we address. We focus on handling *deep dependencies through binary activations*, which we believe to be the crux of the problem. That is why we consider all weights to be real-valued in the present work. An extension to binary weights would be relatively simple, *e.g.* by adding an extra stochastic binarization layer for them.

**SBN Model** Let $x^0$ denote the input to the network (*e.g.* an image to recognize). We define a *stochastic binary network* (SBN) with $L$ layers with neuron outputs $X^{1...L}$ and injected noises $Z^{1...L}$

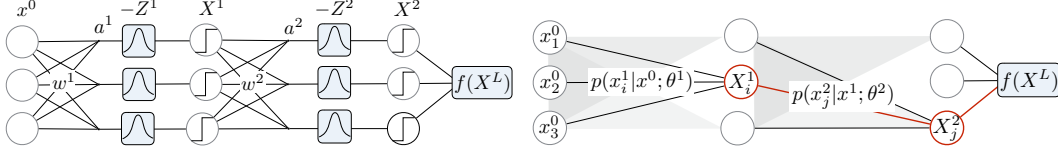

Figure 1: Stochastic binary network with 2 hidden layers. *Left*: latent variable model view (injected noises). *Right*: directed graphical model view (Bayesian network). The PSA method performs explicit summation along paths (highlighted).

as follows (Fig. 1 left):

$$X^0 = x^0; \quad X^k = \text{sgn}(a^k(X^{k-1}; \theta^k) - Z^k); \quad F = f(X^L; \theta^{L+1}). \tag{1}$$

The output $X^k$ of layer $k$ is a vector in $\mathbb{B}^n$, where we denote binary states $\mathbb{B} = \{-1, 1\}$. The network input $x^0$ is assumed real-valued. The noise vectors $Z^k$ consist of $n$ independent variables with a known distribution (*e.g.*, logistic). The network *pre-activation* functions $a^k(X^{k-1}; \theta^k)$ are assumed differentiable in parameters $\theta^k$ and will be typically modeled as affine maps (*e.g.*, fully connected, convolution, concatenation, averaging, *etc.*). Partitioning the parameters $\theta$ by layers as above, incurs no loss of generality since they can in turn be defined as any differentiable mapping $\theta = \theta(\eta)$ and handled by standard backpropagation.

The network *head function* $f(x^L; \theta^{L+1})$ denotes the remainder of the model not containing further binary dependencies. For classification problems we consider the softmax predictive probability model $p(y|x^L; \theta^{L+1}) = \text{softmax}(a^{L+1}(x^L; \theta^{L+1}))$, where the affine transform $a^{L+1}$ computes class scores from the last binary layer. The function $f(X^L; \theta^{L+1})$ is defined as the cross-entropy of the predictive distribution $p(y|x^L; \theta^{L+1})$ relative to the training label distribution $p^*(y|x^0)$.

Due to the injected noises, all states $X$ become random variables and their joint distribution given the input $x^0$ takes the form of a Bayesian network with the following structure (Fig. 1 right):

$$p(x^{1\dots L} \,|\, x^0; \theta) = \prod_{k=1}^{L} p(x^k|x^{k-1}; \theta^k), \quad p(x^k|x^{k-1}; \theta^k) = \prod_{i=1}^{n} p(x_i^k|x^{k-1}; \theta^k). \tag{2a}$$

The equivalence to the injected noise model is established with

$$p(x_j^k = 1|x^{k-1}; \theta^k) = \mathbb{P}\big(a_j^k - Z_j^k > 0\big) = F_Z(a_j^k), \tag{3}$$

where $F_Z$ is the noise cdf. If we consider noises with logistic distribution, $F_Z$ becomes the common sigmoid logistic function and the network with linear pre-activations $a^k(x^{k-1})_j = \sum_j w_{ij}^k x_i^{k-1}$ becomes the well-known *sigmoid belief network* [19].

**Problem** The central problem for this work is to estimate the gradient of the expected loss:

$$\tfrac{\partial}{\partial \theta} \mathbb{E}_Z[F(\theta)] = \tfrac{\partial}{\partial \theta} \sum_{x^{1\dots L}} p(x^{1\dots L}|x^0; \theta) f(x^L; \theta). \tag{4}$$

Observe that when the noise cdf $F_Z$ is smooth, the expected network output $\mathbb{E}_Z[F(\theta)]$ is differentiable in parameters despite having binary activations and a head function possibly non-differentiable in $x^L$. This can be easily seen from the right hand side of (4), where we used the Bayesian network representation of SBN, in which all functions are differentiable in $\theta$. The gradient estimation problem of this kind arises in several learning formulations, please see Appendix A for a motivation of the expected loss objective for training SBNs.

**Bias-Variance Tradeoff** A number of estimators for the gradient (4) exist, both biased and un-biased. Estimators using certain approximations and heuristics have been applied to deep SBNs with remarkable success. The use of approximations however introduces a *systematic error*, *i.e.* these estimators are *biased*. Many theoretical works therefore have focused on development of lower-variance unbiased stochastic estimators, but encounter serious limitations when applied to deep models. At the same time, allowing a small bias may lead to a considerable reduction in variance, and more reliable estimates overall. We advocate this approach and compare the methods using metrics that take into account both the bias and the variance, in particular the mean squared error of the estimator. When the learning has converged to 100% training accuracy (as we will see experimentally for the proposed method) the fact that we used a biased gradient estimator perhaps does not matter.

**Contribution**  The proposed *Path Sample-Analytic* (PSA) method is a biased stochastic estimator. It takes one sample from the model and then applies a series of derandomization and approximation steps. It efficiently approximates the expectation of the stochastic gradient by explicitly computing summations along multiple paths in the network Fig. 1 (right). Such explicit summation over many configurations gives a huge variance reduction, in particular for deep dependencies. The approximation steps needed for keeping the computation simple and tractable, are clearly understood linearizations. They are designed with the goal to obtain a method with the same complexity and structure as the usual backpropagation, including convolutional architectures. This allows to apply the method to deep models and to compute the gradients in parameters of all layers in a single backwards pass.

A second simplification of the method is obtained by further linearizations in PSA and leads to the *Straight-Through* (ST) method. We thus provide the first theoretical justification of straight-through methods for deep models as derived in the SBN framework using a clearly understood linearization and partial summation along paths. This allows to eliminates guesswork and obscurity in the practical application of such estimators as well as opens possibilities for improving them. Both methods perform similar in learning of deep convolutional models, delivering a significantly more stable and better controlled training than preceding techniques.

## 1.1  Related Work

**Unbiased estimators**  A large class of unbiased estimators is based on the REINFORCE [33]. Methods developed to reduce its variance include learnable input-dependent [18] and linearization-based [10] control variates. Advanced variance reduction techniques have been proposed: REBAR [32], RELAX [9], ARM [34]. However the latter methods face difficulties when applied to deep belief models and indeed have never been applied to SBNs with more than two layers. One key difficulty is that they require $L$ passes through the network[1], leading to a quadratic complexity in the number of layers. We compare to ARM, which makes a strong baseline according to comparisons in [34, 9, 32], in a small-problem setting. Previous *direct* comparison of estimator accuracy at the same point was limited to a single neuron setting [34], where PSA would be simply exact. We also compare to MUPROP [10] and variance-reduced REINFORCE, which run in linear time, in the deep setting in Appendix C.3. We observe that the variance of these estimators stays prohibitive for a practical application.

In the case of one hidden layer, our method coincides with several existing unbiased techniques [5, 30, 31]. The RAM estimator [31] can be applied to deep models and stays unbiased but scales quadratically in the number of variables. Our estimator becomes biased for deep models but scales linearly.

**Biased Estimators**  Several works demonstrate successful training of deep networks using biased estimators. One such estimator, based on smoothing the sign function in the SBN model is the *concrete relaxation* [16, 13]. It has been successfully applied for training large-scale SBN in [22]. Methods that propagate moments analytically, known as *assumed density filtering* (ADF), *e.g.*, [26, 8] perform the full analytic approximation of the expectation. ADF has been successfully used in [25].

Many experimentally oriented works successfully apply *straight-through* estimators (STE). Originally considered by Hinton [11] for deep auto-encoders with Bernoulli latent variables and by Bengio et al. [1] for conditional computation, these simple, but not theoretically justified methods were later adopted for training deterministic networks with binary weights and activations [6, 36, 12]. The method simply pretends that the sign function has the derivative of the identity, or of some other function. There have been recent attempts to justify why this works. Yin et al. [35] considers the expected loss over the training data distribution, which is assumed to be Gaussian, and show that in networks with 1 hidden layer the true expected gradient positively correlates with the deterministic STE gradient. Cheng et al. [4] show for networks with 1 hidden layer that STE is approximately related to the projected Wasserstein gradient flow method proposed there. In the case of one hidden layer, Tokui & Sato [31, sec. 6.4] derived STE as a linearization of their RAM estimator for SBN. We derive deep STE in the SBN model by making extra linearizations in our PSA estimator.

## 2 Method

To get a proper view of the problem, we first explain the exact chain rule for the gradient, relating it to the REINFORCE and the proposed method. Throughout this section we will consider the input $x^0$ fixed and omit it from the conditioning such as in $p(x|x^0)$.

### 2.1 Exact Chain Rule

The expected loss in the Bayesian network representation (2) can be written as

$$\mathbb{E}_Z[F] = \sum_{x^1 \in \mathbb{B}^n} p(x^1; \theta^1) \sum_{x^2 \in \mathbb{B}^n} p(x^2 | x^1; \theta^2) \ldots \sum_{x^L \in \mathbb{B}^n} p(x^L | x^{L-1}; \theta^L) f(x^L; \theta^{L+1}) \quad (5)$$

and can be related to the forward-backward marginalization algorithm for Markov chains. Indeed, let $P^k = p(x^k | x^{k-1}; \theta^k)$ be transition probability matrices of size $2^n \times 2^n$ with indices $x^{k-1}$, $x^k$ for $k > 1$ and $P^1 = p(x^1; \theta^1)$ be the row vector of size $2^n$. Then (5) is simply the product of these matrices and the vector $f$ of size $2^n$. The computation of the gradient is as "easy", *e.g.* for the gradient in $\theta^1$ we have:

$$g^1 := \tfrac{\partial}{\partial \theta^1} \mathbb{E}_Z[F] = D^1 P^2 P^3 \ldots P^L f, \quad (6)$$

where $D^1$ is the size $\dim(\theta^1) \times 2^n$ transposed Jacobian $\frac{\partial p(x^1; \theta^1)}{\partial \theta^1}$. Expression (6) is a product of transposed Jacobians of a deep model. Multiplying them in the right-to-left order requires only matrix-vector products and it is a back-propagation algorithm computing the exact gradient. As impractical as it may be, it is still much more efficient than the brute force enumeration of all $2^{nL}$ joint configurations.

The well-known REINFORCE [33] method replaces the intractable summation over $x$ with sampling $x$ from $p(x^{1\ldots L}; \theta)$ and uses the stochastic estimate

$$\tfrac{\partial}{\partial \theta^1} \mathbb{E}_Z[F] \approx \tfrac{\partial p(x^1; \theta^1)}{\partial \theta^1} \tfrac{f(x^L)}{p^1(x^1; \theta^1)}. \quad (7)$$

While this is cheap to compute (and unbiased), it utilizes neither the shape of the function $f$ beyond its value at the sample nor the dependence structure of the model.

### 2.2 PSA Algorithm

We present our method for a deep model. The single layer case is a special case which is well covered in the literature [31, 30, 5] and is discussed in Lemma B.1. Let us consider the gradient in parameters $\theta^l$ of layer $l$. Starting from the RHS of (4), we can move derivative under the sum and directly differentiate the product distribution (2):

$$g^l := \tfrac{\partial}{\partial \theta^l} \sum_x p(x; \theta) f(x^L; \theta^{L+1}) = \sum_x f(x^L) \tfrac{\partial}{\partial \theta^l} p(x; \theta) = \sum_x \tfrac{p(x) f(x^L)}{p(x^l | x^{l-1})} \tfrac{\partial}{\partial \theta^l} p(x^l | x^{l-1}; \theta^l), \quad (8)$$

where the dependence on $\theta$ in $p$ and $f$ is omitted once it is outside of the derivative. The fraction $\frac{p(x)}{p(x^l | x^{l-1})}$ is a convenient way to write the product $\prod_{k=1 | k \neq l}^L p(k^k | x^{k-1})$, *i.e.* with factor $l$ excluded.

At this point expression (8) is the exact chain rule completely analogous to (6). A tractable back propagation approximation is obtained as follows. Since $p(x^l | x^{l-1}; \theta^l) = \prod_{i=1}^n p(x_i^l | x^{l-1}; \theta^l)$, its derivative in (8) results in a sum over units $i$:

$$g^l = \sum_x \sum_i \tfrac{p(x)}{p(x_i^l | x^{l-1})} D_i^l(x) f(x^L), \qquad \text{where } D_i^l(x) = \tfrac{\partial}{\partial \theta^l} p(x_i^l | x^{l-1}; \theta^l). \quad (9)$$

We will apply a technique called *derandomization* or Rao-Blackwellization [20, ch. 8.7] to $x_i^l$ in each summand $i$. Put simply, we sum over the two states of $x_i^l$ explicitly. The summation computes a portion of the total expectation in a closed form and as such, naturally and guaranteed, reduces the variance. The estimator with this derandomization step is an instance of the general *local expectation gradient* [30]. The derandomization results in the occurrence of the differences of products:

$$\prod_j p(x_j^{l+1} | x_{\downarrow i}^l) - \prod_j p(x_j^{l+1} | x^l), \quad (10)$$

where $x_{\downarrow i}^l$ denotes the state vector in layer $l$ with the sign of unit $i$ flipped. We approximate this difference of products by *linearizing* it (making a 1st order Taylor approximation) in the differences of its factor probabilities, *i.e.*, replacing the difference (10) with

$$\sum_j \prod_{j' \neq j} p(x_{j'}^{k+1} | x^k) \Delta_{i,j}^{k+1}(x), \quad \text{where } \Delta_{i,j}^{k+1}(x) = p(x_j^{k+1} | x^k) - p(x_j^{k+1} | x_{\downarrow i}^k). \quad (11)$$

The approximation is sensible when $\Delta_{i,j}^{k+1}$ are small. This holds *e.g.*, in the case when the model has many units in layer $k$ that all contribute to preactivation so that the effect of flipping a single $x_i^k$ is small.

Notice that the approximation results in a sum over units $j$ in the next layer $l+1$, which allows us to isolate a summand $j$ and derandomize $x_j^{l+1}$ in turn. Chaining these two steps, derandomization and linearization, from layer $l$ onward to the head function, we obtain summations over units in all layers $l \ldots L$, equivalent to considering all paths in the network, and derandomize over all binary states along each such path (see Fig. 1 right). The resulting approximation of the gradient $g^l$ gives

$$\tilde{g}^l = \sum_x p(x) D^l(x) \Delta^{l+1}(x) \cdots \Delta^L(x) df(x) =: \sum_x p(x) \hat{g}^l(x), \tag{12}$$

where $D^l$, defined in (9), is the Jacobian$^\mathsf{T}$ of layer probabilities in parameters (a matrix of size $\dim(\theta^l) \times n$); $\Delta_{i,j}^k$ defined in (11) are $n \times n$ matrices, which we call *discrete Jacobians*$^\mathsf{T}$; and $df$ is a column vector with coordinates $f(x^L) - f(x_{\downarrow i}^L)$, *i.e.* a *discrete gradient* of $f$. Thus a stochastic estimate $\hat{g}^l(x)$ is a product of Jacobians$^\mathsf{T}$ in the ordinary activation space and can therefore be conveniently computed by back-propagation, *i.e.* multiplying the factors in (12) from right to left.

This construction allows us to approximate a part of the chain rule for the Markov chain with $2^n$ states as a chain rule in a tractable space and to do the rest via sampling. This way we achieve a significant reduction in variance with a tractable computational effort. This computation effort is indeed optimal is the sense that it matches the complexity of the standard back-propagation, which matches the complexity of forward propagation alone, *i.e.*, the cost of obtaining a sample $x \sim p(x)$.

**Derivation** The formal construction is inductive on the layers. Consider the general case $l < L$. We start from the expression (9) and apply to it Proposition 1 below recurrently, starting with $J^l = D^l$. Matrices $J^k$ will have an interpretation of composite Jacobians$^\mathsf{T}$ from layer $l$ to layer $k$.

**Proposition 1.** Let $J_i^k(x)$ be functions that depend only on $x^{1 \ldots k}$ and are *odd* in $x_i^k$: $J_i^k(x^k) = -J_i^k(x_{\downarrow i}^k)$ for all $i$. Then

$$\sum_x p(x) \sum_i \frac{J_i^k(x) f(x^L)}{p(x_i^k | x^{k-1})} \approx \sum_x p(x) \sum_j \frac{J_j^{k+1}(x) f(x^L)}{p(x_j^{k+1} | x^k)}, \quad \text{where } J_j^{k+1} = \sum_i J_i^k(x) \Delta_{i,j}^{k+1}(x) \tag{13}$$

and the approximation made is the linearization (11). Functions $J_j^{k+1}$ are odd in $x_j^{k+1}$ for all $j$.

The structure of (13) shows that we will obtain an expression of the same form but with the dividing probability factor from the next layer, which allows to apply it inductively. To verify the assumptions at the induction basis, observe that according to (3), $D_i^l(x) = \frac{\partial}{\partial \theta^l} p(x_i^l | x^{l-1}; \theta^l) = p_Z(a_i^l) x_i^l \frac{\partial}{\partial \theta^l} a_i^l(x^{l-1}; \theta^l)$, hence it depends only on $x^{l-1}$, $x^l$ and is odd in $x_i^l$.

In the last layer, the difference of head functions occurs instead of the difference of products. Thus instead of (13), we obtain for $k = L$, without an approximation, the expression

$$\sum_x p(x) \sum_i \frac{J_i^L(x) f(x^L)}{p(x_i^L | x^{L-1})} = \sum_x \sum_i p(x) J_i^L(x) df_i(x). \tag{14}$$

Applying Proposition 1 inductively, we see that the initial $J^l$ is multiplied with a discrete Jacobian$^\mathsf{T}$ $\Delta^{k+1}$ on each step $k$ and finally with $df$. Note that neither matrices $\Delta^k$ nor $df$ depend on the layer we started from. The final result of inductive application of Proposition 1 is exactly the expected matrix product (12). The key for computing a one-sample estimate $\hat{g}(x)^l = D^l \Delta^{l+1} \cdots \Delta^L df$ is to perform the multiplication from right to left, which only requires matrix-vector products. Observe also that the part $\Delta^k \cdots \Delta^L df$ is common for our approximate derivatives in all layers $l < k$ and therefore needs to be evaluated only once.

**Algorithm** Since backpropagation is now commonly automated, we opt to define forward propagation rules such that their automatic differentiation computes what we need. The algorithm in this form is presented in Algorithm 1. First, we have substituted the noise model (3) to compute $\Delta_{i,j}^k$ as in Line 6. The `detach` method available in PyTorch [21] obtains the value of the tensor but excludes it from back-propagation. It is applied to $\Delta^k$ since it is already a Jacobian$^\mathsf{T}$ and we do not want to differentiate it. The recurrent computation of $q^k$ in Line 7 serves to generate the computation of $\hat{g}^l(x)$ on the backward pass. Indeed, variables $q^k$ always store just zero values as ensured by Line 8 but

**Algorithm 1:** Path Sample-Analytic (**PSA**)

**Input:** Network parameters $\theta$, input $x^0$
**Output:** The expression $E$ generating the derivative
1  Initialize: $q^0 = 0$;
2  **for** *layer k with Bernoulli output* **do**
3  $\quad$ $a_j^k = a_j^k(x^{k-1}; \theta^k)$;
4  $\quad$ Sample layer output state $x_j^k \in \{-1, 1\}$ with probability of 1 given by $F_Z(a_j^k)$;
5  $\quad$ Compute discrete Jacobians$^\mathsf{T}$:
6  $\quad$ $\Delta_{i,j}^k = \text{detach}(x_j^k(F_Z(a_j^k) - F_Z(a_j^k(x_{\downarrow i}^{k-1}; \theta^k))))$;
7  $\quad$ Generate chain dependence:
8  $\quad$ $q_j^k = x_j^k F_Z(a_j^k) + \sum_i \Delta_{i,j}^k q_i^{k-1}$;
9  $\quad$ $q^k := q^k - \text{detach}(q^k)$;
10  Last layer:
11  $E = f(x^L; \theta^{L+1}) + \sum_i \text{detach}(f(x^L) - f(x_{\downarrow i}^L))q_i^L$;
12  **return** $E$

**Algorithm 2:** Straight-Through (**ST**)

**Input:** Network parameters $\theta$, input $x^0$
**Output:** The expression $E$ generating the derivative
1  **for** *layer k with Bernoulli output* **do**
2  $\quad$ $a_j^k = a_j^k(x^{k-1}; \theta^k)$;
3  $\quad$ Sample layer output states $x_j^k \in \{-1, 1\}$ with probability of 1 given by $F_Z(a_j^k)$;
4  $\quad$ Compute $\tilde{x}_j^k = 2F_Z(a_j^k)$;
5  $\quad$ Binary state with a derivative generator: $x^k := x^k + \tilde{x}^k - \text{detach}(\tilde{x}^k)$;
6  Last layer's output is the derivative generator:
7  **return** $E = f(x^L)$

have non-zero Jacobians as needed. A similar construct is applied for the final layer. It is easy to see that differentiating the output $E$ w.r.t. $\theta^l$ recovers exactly $\hat{g}(x)^l = D^l \Delta^{l+1} \cdots \Delta^L df$.

This form is simpler to present and can serve as a reference. The large-scale implementation detailed in Appendix B.5 defines custom backward operations, avoiding the overhead of computing variables $q^k$ in the forward pass. The overhead however is a constant factor and the following claim applies.

**Proposition 2.** The computation complexity of gradient by Algorithm 1 in all parameters of a network with convolutional and fully connected layers is the same as that of the standard back-propagation.

The proof is constructive in that we specify algorithms how the required computation for all flips as presented in Algorithm 1 can be implemented with the same complexity as standard back-propagation. Moreover, in Appendix B.5 we show how to implement transposed convolutions for the case of logistic noise to achieve the FLOPs complexity as low as 2x standard back-propagation.

Since the steps we perform for the last layer in (14) do not involve linearization, the estimate of the gradient in $\theta_L$ is always unbiased. Furthermore, if the product (10) has only one factor, the approximation (11) is exact and we have the following desirable property for this special case:

**Proposition 3.** Algorithm 1 is an unbiased gradient estimator for networks with only one unit in layers $1 \dots L-1$ and any number of units in layer $L$.

## 2.3 The Straight-Through Estimator

**Proposition 4.** Assume that pre-activations $a^k(x^{k-1}; \theta)$ are multilinear functions in the binary inputs $x^{k-1}$, and the objective $f$ is differentiable. Then, approximating $F_Z$ and $f$ linearly around their arguments for the sampled base state $x$ in Algorithm 1, we obtain the straight-through estimator in Algorithm 2.

By applying the stated linear approximations, the proof of this proposition shows that the necessary derivatives and Jacobians in Algorithm 1 can be formally obtained as derivatives of the noise cdf $F_Z$ w.r.t. parameters $\theta$ and the binary states $x^{k-1}$. Despite not improving the theoretical computational complexity compared to PSA, the implementation is much more straightforward. We indeed see that Algorithm 2 belongs to the family of straight-through estimators, using hard threshold activation in the forward pass and a smooth substitute for the derivative in the backward pass. The key difference to many empirical variants being that it is clearly paired with the SBN model and the choice of the smooth function for the derivative matches the model and the sampling scheme. As we have discovered later on, it matches exactly to the original description of such straight-through by Hinton [11] (with the difference that we use $\pm 1$ encoding and thus scaling factors 2 occur).

In the case of logistic noise, we get $2F_Z(a) = 2\sigma(a) = 1 + \tanh(a/2)$, where the added 1 does not affect derivatives. This recovers the popular use of $\tanh$ as a smooth replacement, however, the slope of the function in our case must match the sampling distribution. In the experiments we

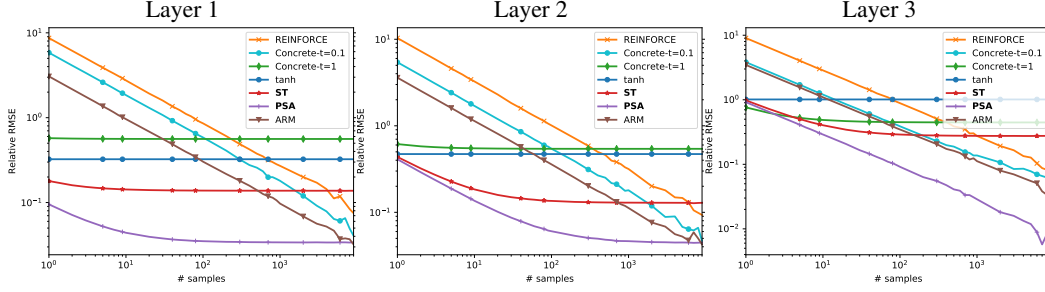

Figure 2: Root mean squared error of the gradient in layers 1 to 3 relative to the true gradient length after epoch 1 of training with REINFORCE. Layer 1 parameters correspond to $\theta^1$ in (1) – weights and biases defining preactivations of layer 1 Bernoulli states. Unbiased estimators always improve with more samples. Biased estimators only improve up to a point. However, biased methods may be more accurate when using fewer samples and the discrepancy significantly increases with layer depth.

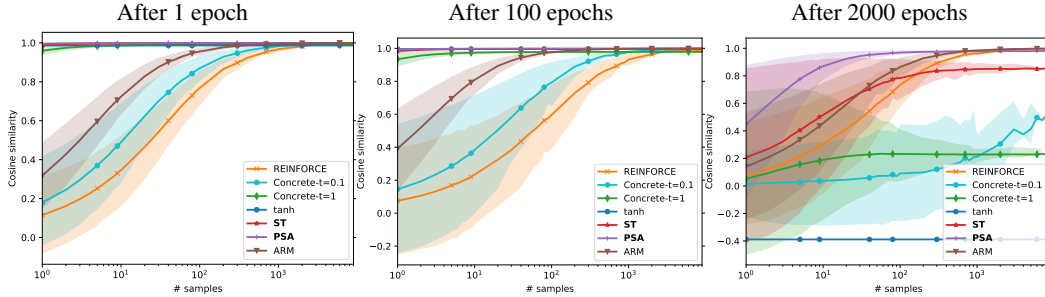

Figure 3: Cosine similarity of the estimated gradient to the true gradient in layer 1 at different points during training. The lines show the mean of the cosine similarity of the $N$-sample estimator. The shaded areas shows the interval containing 70% of the trials, illustrating the scatter of values that can be obtained in a random trial. It is seen that for some estimators there are good chances of failing to produce a positive cosine, *i.e.* a valid descent direction.

compare it to a popular straight-through variant (*e.g.*, [12]) where the gradient of *hard tanh* function, $\min(\max(x, -1), 1)$ is taken. The mismatch between the SBN model and the gradient function in this case leads to a severe loss of gradient estimation accuracy. The proposed derivation thus eliminates lots of guesswork related to the use of such estimators in practice, allows to assess the approximation quality and understand its limitations.

## 3 Experiments

We evaluate the quality of gradient estimation on a small-scale dataset and the performance in learning on a real dataset. In both cases we use SBN models with logistic noises, which is a popular choice. Despite the algorithm and the theory is applicable for any continuous noise distribution, we have optimized the implementation of convolutions specifically for logistic noises.

**Gradient Estimation Accuracy**    To evaluate the accuracy of gradient estimators, we implemented the exact method, feasible for small models. We use the simple problem instance shown in Fig. B.1(a) with 2 classes and 100 training points per class in 2D and a classification network with 5-5-5 Bernoulli units. To study the bias-variance tradeoff we vary the number of samples used for an estimate and measure the Mean Squared Error (MSE). For unbiased estimators using $N$ samples leads to a straightforward variance reduction by $1/N$, visible in the log-log plot in Fig. 2 as straight lines. To investigate how the gradient quality changes with the layer depth we measure Root MSE in each of the 3 layers separately. It is seen in Fig. 2 that the proposed PSA method has a bias, which is the asymptotic value of RMSE when increasing the number of samples. However, its RMSE accuracy with 1 sample is indeed not worse than that of the advanced unbiased ARM method with $10^3$ samples. We should note that in more deterministic points, where the network converges to during the training, the conditions for approximation hold less well and the bias of PSA may become more significant while unbiased methods become more accurate (but the gradient signal diminishes). Fig. 2

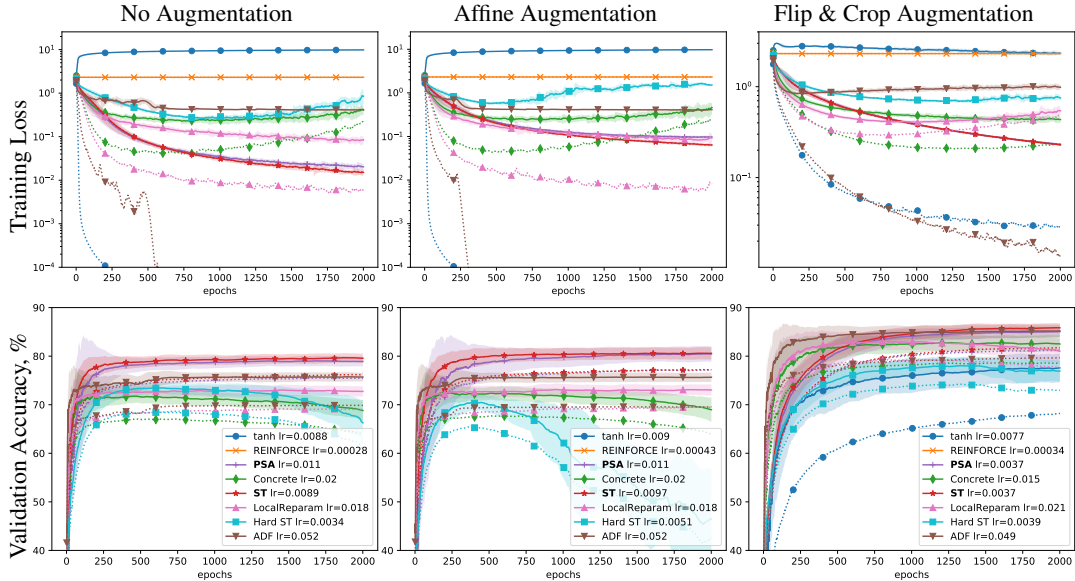

| | |
|---|---|
| REINFORCE | Unbiased estimator [33]. |
| Tanh | Replace $\text{sign}(a - Z)$ by $E_Z[\text{sign}(a - Z)] = \tanh(a/2)$. |
| Concrete-$t$ | Concrete Relaxation [16] with the relaxation parameter $t$. |
| HardST | STE with the gradient of clamped identity, $\max(\min(a, 1), -1)$. |
| ADF | Assumed density filtering, *e.g.*, [26], the equivalent of PBNET method in [22] for real weights. |
| LocalReparam | Approximating pre-activations with normal distribution and sampling them. |

Figure 4: Learning comparison on CIFAR-10. Solid loss curves measure the SBN expected loss. Doted loss curves indicate the relaxed objectives used by respective methods (where applicable). Solid accuracy curves are using 10-sample expected predictive probabilities of SBN and dotted curves only 1-sample predictive probabilities. All curves are smoothed over iterations and shaded areas denote $3\times$std w.r.t. smoothing. The automatically found learning rates are displayed in the legend.

also confirms experimentally that PSA is always more accurate than ST and has no bias in the last hidden layer, as expected. Additional experiments studying the dependence of accuracy on network width, depth and comparison with more unbiased baselines are given in Appendix C (Figs. C.3, C.5 and C.6). Both PSA and ST methods are found to be in advantage when increasing depth and width.

The cosine similarity metric measured in Fig. 3 is more directly relevant for optimization. If it is close to one, we have an accurate gradient direction estimate. If it is positive, we still have a descent direction. Negative cosine similarity will seriously harm the optimization. For this evaluation we take the model parameters at epochs 1, 100 and 2000 of a reference training and measure the cosine similarity of gradients in layer 1. We see that methods with high bias may systematically fail to produce a descent direction while methods with high variance may produce wrong directions too often. Both effects can slow down the optimization or steer it wrongly.

The proposed PSA method achieves the best accuracy in the practically important low-sample regime. The ST method is often found inferior and we know why: the extra linearization does not hold well when there are only few units in each layer. We expect it to be more accurate in larger models.

**Deep Learning** To test the proposed methods in a realistic learning setting we use CIFAR-10 dataset and network with 8 convolutional and 1 fully connected layers (Appendix C). The first and foremost purpose of the experiment is to assess how the methods can optimize the training loss. We thus avoid using batch normalization, max-pooling and huge fully connected layers. When comparing to a number of existing techniques, we find it infeasible to optimize all hyper-parameters such as learning rate, schedule, momentum *etc.* per method by cross-validation. However, compared methods significantly differ in variance and may require significantly different parameters. We opt to use SGD with momentum with a constant learning rate found by an automated search per method (Appendix C). While this may be suboptimal, it nevertheless allows to compare the behavior of algorithms and analyze the failure cases. We further rely on SGD to average out noisy gradients with a suitable

learning rate and therefore use 1-sample gradient estimates. To modulate the problem difficulty, we evaluate 3 data augmentation scenarios: no augmentation, affine augmentation, Flip&Crop augmentation.

Fig. 4 and Fig. C.1 show the training performance of evaluated methods. Both PSA and ST achieve very similar and stable training performance. This verifies that the extra linearization in ST has no negative impact on the approximation quality in large models. For methods Tanh, Concrete, ADF, LocalReparam we can measure their relaxed objectives, whose gradients are used for the training (*e.g.* the loss of a standard neural network with tanh activations for Tanh). The training loss plots reveal a significant gap between these relaxed objectives and the expected loss of the SBN. While the relaxed objectives are optimized with an excellent performance, the real objective stalls or starts growing. This agrees with our findings in Fig. 3 that biased methods may fail to provide descent directions. HardST method, seemingly similar to our ST, performs rather poorly. Despite its learning rate is smaller than that of ST, it diverges in the first two augmentation cases, presumably due to wrongly estimated gradient directions. As we know from preceding work, good results with these existing methods are possible, in particular we also see that ADF with Flip&Crop augmentation achieves very good validation accuracy despite poor losses. We argue that in methods where bias may become high there is no sufficient control of what the optimization is doing and one needs to balance with empirical guessing. Finally, the REINFORCE method requires a very small learning rate in order not to diverge and the learning rate is indeed so small that we do not see the progress. We investigate if further by attempting input-dependent baselines in Appendix C.3. While it can optimize the objective in principle, the learning rate stays very small.

Please see further details on implementation and the training setup in Appendices B.5 and C. The implementation is available at `https://github.com/shekhovt/PSA-Neurips2020`.

## 4    Conclusion

We proposed a new method for estimating gradients in SBNs, which combines an approximation by linearization and a variance reduction by summation along paths, both clearly interpretable. We experimentally verified that our PSA method has a practical bias-variance tradeoff in the quality of the estimate, runs with a reasonable constant factor overhead, as compared to standard back propagation, and can improve the learning performance and stability. The ST estimator obtained from PSA gives the first theoretical justification of straight-through methods for deep models, opening the way for their more reliable and understandable use and improvements. Its main advantage is implementation simplicity. While the estimation accuracy may suffer in small models, we have observed that it performs on par with PSA in learning large models, which is very encouraging. However, the influence of the systematic bias on the training is not fully clear to us, we hope to study it further in the future work.

## Broader Impact

The work promotes stochastic binary networks and improves the understanding and efficiency of training methods. We therefore believe ethical concerns are not applicable. At the same time, developing more efficient training methods for binary networks, we believe may further increase the researchers and engineers interest in low-energy binary computations and aid progress in embedded applications such as speech recognition and vision. In the field of stochastic computing, which is rather detached at the moment, the stochasticity is treated as a source of errors and accumulators are used in every layer just to mimic smooth activation function [14, 15]. It appears to us that when stochastic binary computations are made useful instead, the related hardware designs can be made more efficient and stable w.r.t. to errors.

## Acknowledgments and Disclosure of Funding

We thank the anonymous peers for pointing out related work and helping to clarify the presentation. We gratefully acknowledge our funding agencies. A.S. was supported by the project "International Mobility of Researchers MSCA-IF II at CTU in Prague" (CZ.02.2.69/0.0/0.0/18_070/0010457). V.Y.

was supported by Samsung Research, Samsung Electronics. B.F. was supported by the Czech Science Foundation grant no. 19-09967S.

## Footnotes

[1]See REBAR section 3.3, RELAX section 3.2, ARM Alg. 2. Different propositions to overcome the complexity limitation appear in appendices of several works, including also [5], which however have not been tested in deep models.

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
