[Supplementary Material]

# Path Sample-Analytic Gradient Estimators
# for Stochastic Binary Networks (Appendix)

## Contents

## A  Learning Formulations

Here we give a clarification on the learning formulation used in this work. During the training we consider the expected loss of a randomized predictor:

$$\mathbb{E}_{(x^0,y)}\mathbb{E}_Z\big[F(\theta)\big] = -\mathbb{E}_{(x^0,y)}\mathbb{E}_Z\big[\log p(y|X^L;\theta)\big], \qquad (15)$$

where $(x^0, y) \sim \texttt{data}$. However at the test time we evaluate the expected predictor $\mathbb{E}_Z\big[p(y|X^L;\theta)\big]$, considering $Z$ as *latent* variables, which can be interpreted as an *ensemble* of binary networks (see Fig. B.1 c-f). The loss of this marginal predictor would be rather given by:

$$-\mathbb{E}_{(x^0,y)}\big[\log\mathbb{E}_Z\big[p(y|X^L;\theta)\big]\big]. \qquad (16)$$

This setup is similar to dropout [27] with latent multiplicative Bernoulli noises. The expected loss (15) upper bounds the marginal loss (16) (by Jensen's inequality), so that minimizing it also minimizes (16). However, unlike with dropout, the following observation holds for SBN models.

**Proposition A.1.** In the family of models (1) with free scale and bias in all coordinates of $a^k$ there is always an effectively deterministic model (with no injected noises) that achieves the same or better expected loss (15).

This means that the model will tend to be deterministic and fit the classification boundary but not the data ambiguity (see Fig. B.1 b,c). Being aware of this, we note that it is nevertheless a common way to train classification models (*e.g.*, dropout). Furthermore, the upper bound may be tightened by considering a multi-sample bound [29, 23] or variational bounds (applied for shallow models in [34, 9]). These extensions are left for future work as they require the ability to estimate gradients of the expected loss (15) in the first place. We can nevertheless see from the example in Fig. B.1 that the SBN family can be expressive when trained appropriately.

## B  Proofs

This section contains proofs, technical details and extended discussion that did not fit in the main paper.

### B.1  Training with Expected Loss tends to Deterministic Models

**Proposition A.1.** In the family of models (1) with free scale and bias in all coordinates of $a^k$ there is always an effectively deterministic model (with no injected noises) that achieves the same or better expected loss (15).

Raiko et al. [23] give a related theorem, but do not show the preferred deterministic strategy to be realizable in the model family.

*Proof.* Let $\theta$ be parameters of the model optimizing (15). Let then $z^*$ be a maximizer of $\mathbb{E}_{(x^0,y*)\sim\text{data}}\big[f(x^L(z), y; \theta)\big]$. Consider the case of a linear layer $a(x) = W^Tx + b$ with the output

Figure B.1: Example problem to classify points in 2D with overlapping distributions. (a) Data points. (b) Classification model trained with the expected loss (15): the optimal solution tends to deterministic prediction. (c) Same model trained with a 10-sample bound [23], closer to the marginal likelihood (16). The model fits the uncertainty of the data. (d-f) Examples of the ensemble members obtained by fixing a particular realization of the noise variables $Z$ in all layer for the model in (c).

$\mathrm{sgn}(W^T x + b - Z)$. Chose as new parameters $W' = sW$, $b' = s(b - z^*)$ for $s \to \infty$. Since $Z$ has a finite variance, this ensures that $\mathrm{sgn}(W'^T X + b' - Z) = \mathrm{sgn}(W^T X + b - z^* - Z/s) \to \mathrm{sgn}(W^T X + b - z^*)$. The network with new weights is deterministic as it efficiently scales all noises to zero and it achieves same or better expected loss. The same argument applies whenever $a$ has a free scale and bias degrees of freedom. $\square$

Let us remark that the conditions are not met in the following cases:

- The pre-activation does not have some degree of freedom, *e.g.*, there is no bias term. This case is obvious.
- Pre-activations of different outputs do not have independent degrees of freedom per output. *e.g.*, in a convolutional network we can suppress all the noises by scaling them down, however since the noises of all pre-activations are independent (not spatially identical), we cannot represent the bias from $z^*$ with the convolution bias which is spatially homogenous.
- The network uses parameter sharing in some other way, *e.g.*, a Siamese network for matching.

These exceptions actually imply that training with expected loss a convolutional network in Section 3 tends to be deterministic but will not collapse to a fully deterministic state as it is suboptimal.

## B.2 PSA Derivation and Properties

**Proposition 1.** Let $J_i^k(x)$ be functions that depend only on $x^{1...k}$ and are *odd* in $x_i^k$: $J_i^k(x^k) = -J_i^k(x_{\downarrow i}^k)$ for all $i$. Then

$$\sum_x p(x) \sum_i \frac{J_i^k(x) f(x^L)}{p(x_i^k | x^{k-1})} \approx \sum_x p(x) \sum_j \frac{J_j^{k+1}(x) f(x^L)}{p(x_j^{k+1} | x^k)}, \quad \text{where } J_j^{k+1} = \sum_i J_i^k(x) \Delta_{i,j}^{k+1}(x) \quad (13)$$

and the approximation made is the linearization (11). Functions $J_j^{k+1}$ are odd in $x_j^{k+1}$ for all $j$.

*Proof.* Starting from LHS of (13) we take the sum in $x_i^k$ explicitly. The factors involving $x_i^k$ (after cancellation of the denominator with the respective term in $p(x)$) are

$$p(x^{k+1} | x^k) J_i^k(x^k), \quad (17)$$

where we omit the dependance of $J_i^k$ on $x^{1...k-1}$, not relevant for the sum in $x_i^k$. Using the oddness of $J_i^k$, the sum of (17) in $x_i^k$ can be written as

$$p(x^{k+1} | x^k) J_i^k(x^k) + p(x^{k+1} | x_{\downarrow i}^k) J_i^k(x_{\downarrow i}^k) = \left( p(x^{k+1} | x^k) - p(x^{k+1} | x_{\downarrow i}^k) \right) J_i^k(x^k). \quad (18)$$

Though this expression formally depends on $x_i^k$, it is by design invariant to $x_i^k$. Thus $x_i^k$ has been *derandomized*. We multiply (18) with $1 = \sum_{x_i^k} p(x_i^k | x^{k-1})$ to obtain

$$\sum_{x_i^k} p(x_i^k | x^{k-1}) \left( p(x^{k+1} | x^k) - p(x^{k+1} | x_{\downarrow i}^k) \right) J_i^k(x^k), \quad (19)$$

which allows to put this expression back as a part of the joint sum in $x$ in (13). We thus obtain in (13):

$$\sum_{i,x} p(x^{1...k}) p(x^{k+2...L} | x^{k+1}) \left( p(x^{k+1} | x^k) - p(x^{k+1} | x_{\downarrow i}^k) \right) J_i^k(x) f(x^L). \quad (20)$$

Recalling that $p(x^{k+1} | x^k) = \prod_j p(x_j^{k+1} | x^k)$, the product linearization (11) gives

$$p(x^{k+1} | x^k) - p(x^{k+1} | x_{\downarrow i}^k) \approx \sum_j \frac{p(x^{k+1} | x^k)}{p(x_j^{k+1} | x^k)} \Delta_{i,j}^{k+1}(x), \quad (21)$$

where the division is used to represent the factor that needs to be excluded. Substituting this into (20) we get the resulting expression in (13). Finally, $\Delta_{i,j}^{k+1}(x)$ is odd in $x_j^{k+1}$ and $J^k(x)$ does not depend on $x_j^{k+1}$, and therefore $J_j^{k+1}$ is odd in $x_j^{k+1}$. $\square$

**Case $l=L$**  We now prove the expression (14) claimed as the derandomization result for the last layer when propagating $J^L$. Let us consider gradient in parameters of the last layer, $g^L$. In this case, the gradient expression (9) becomes

$$\sum_{x^{1\ldots L-1}} p(x) \sum_i \sum_{x^L} \frac{d_i^L(x)}{p(x^L|x^{L-1})} f(x^L). \tag{22}$$

Then derandomization over $x^L$ takes a particular simple form, which can be described by the following standalone lemma.

**Lemma B.1.** Let $X_i$ be independent $\mathbb{B}$-valued Bernoulli with probability $p(x_i; \theta)$ for $i = 1 \ldots n$ and $f \colon \mathbb{B}^n \to \mathbb{R}$. Let $x$ be a joint sample and $x_{\downarrow i}$ denote the joint state with $x_i$ flipped. Then

$$\sum_i \tfrac{\partial}{\partial \theta} p(x_i; \theta)\big(f(x) - f(x_{\downarrow i})\big) \tag{23}$$

is an unbiased estimate of $\tfrac{\partial}{\partial \theta} \sum_x p(x; \theta) f(x)$.

Analogous results exist in the literature, *e.g.* [5] considers general discrete and continuous distributions.

*Proof.* We differentiate the product of probabilities in the expectation:

$$\tfrac{\partial}{\partial \theta} \sum_x \prod_i p(x_i; \theta) f(x) = \sum_x \sum_i \tfrac{p(x)}{p(x_i)} \tfrac{\partial}{\partial \theta} p(x_i; \theta) f(x). \tag{24}$$

We then compute the sum over $x_i$ for each summand $i$ explicitly, obtaining

$$\sum_i \sum_{x_{\neg i}} p(x_{\neg i}) \tfrac{\partial}{\partial \theta} \Big( p(x_i; \theta) f(x) + (1 - p(x_i; \theta)) f(x_{\downarrow i}) \Big),$$

where $x_{\neg i}$ denotes excluding the component $i$. Since the expression in the brackets is invariant of $x_i$, we multiply by the factor $1 = \sum_{x_i} p(x_i)$ and get

$$\sum_x p(x) \sum_i \big(f(x) - f(x_{\downarrow i})\big) \tfrac{\partial}{\partial \theta} p(x_i). \tag{25}$$

Thus (23) is a single sample unbiased estimate of (25). $\qquad\square$

**Proposition 3.** Algorithm 1 is an unbiased gradient estimator for networks with only one unit in layers $1 \ldots L-1$ and any number of units in layer $L$.

*Proof.* The product linearization is not used anywhere in the method when we have a single binary unit in each hidden layer with $l < L$, nor it is used in the last layer. We therefore make no approximations and the 1-sample estimate is unbiased. $\qquad\square$

### B.3   Last Layer Enhancement

In Algorithm 1 Line 10 we defined $E$ so that the gradient in $\theta^{L+1}$ is the common stochastic estimate $\frac{\partial f(x^L; \theta^{L+1})}{\partial \theta^{L+1}}$. We now propose an improvement to this estimate. Intuitively, we want to utilize the values $f(x_{\downarrow i}^L; \theta^{L+1})$ for all $i$ that we compute anyway.

Estimating $\sum_x p(x) \tfrac{\partial}{\partial \theta} f(x^L; \theta)$ means to estimate the expected value of the function $g(x^L) = \tfrac{\partial}{\partial \theta} f(x^L; \theta)$, without further derivatives involved. We have the following lemma that applies derandomization over units in the last layer.

**Lemma B.2.** Let $X_i$ be independent $\mathbb{B}$-valued Bernoulli with probability $p(x_i)$ for $i = 1 \ldots n$ and $g \colon \mathbb{B}^n \to \mathbb{R}$. Let $x$ be a joint sample. Then

$$g(x) + \gamma \sum_i \big(g(x_{\downarrow i}) - g(x)\big)(1 - p(x_i)) \tag{26}$$

is an unbiased estimate of $\mathbb{E}_X[g(X)]$.

*Proof.* We expand for some fixed $i$:

$$\mathbb{E}_X[g] = \sum_x p(x) g(x) = \sum_{x_{\neg i}} p(x_{\neg i}) \sum_{x_i} p(x_i) g(x) \tag{27}$$

$$= \sum_{x_{\neg i}} p(x_{\neg i}) \Big( p(x_i) g(x) + p(-x_i) g(x_{\downarrow i}) \Big). \tag{28}$$

Observe that the bracket does not depend on $x_i$. We can therefore rewrite the expression as

$$\sum_x p(x) \Big( p(x_i) g(x) + p(-x_i) g(x_{\downarrow i}) \Big) \tag{29}$$

$$= \sum_x p(x) \Big( (1 - p(-x_i)) g(x) + p(-x_i) g(x_{\downarrow i}) \Big) \tag{30}$$

$$= \sum_x p(x) \Big[ g(x) + (g(x_{\downarrow i}) - g(x)) p(-x_i) \Big]. \tag{31}$$

This shows that

$$\sum_x p(x)\big(g(x_{\downarrow i}) - g(x)\big)p(-x_i) = 0. \tag{32}$$

So it serves as a variance reduction baseline. Moreover, if we have access to the two values $g(x_{\downarrow i})$ and $g(x)$, it is the perfect baseline, as adding it results in the complete sum in $x_i$. Taking the average over $i$, *i.e.* choosing $\gamma = \frac{1}{n}$ gives an estimate with a decreased variance, however it is not straightforward which value of $\gamma$ gives the best variance reduction as estimates (32) for all $i$ are not independent. $\qquad\square$

Setting $\gamma = \frac{1}{n}$ is a natural choice that guarantees a reduction in variance. This improvement can be implemented as a simple replacement of the last line of the algorithm to:

$$\pi_i^L = 1 - \text{detach}((x_i^L + 1)/2 + x_i^L F_Z(a_i^L)) \tag{33a}$$

$$E = f(x^L; \theta) + \sum_i (f(x^L; \theta) - f(x_{\downarrow i}^L; \theta))(q_i^L - \tfrac{\pi_i^L}{n}). \tag{33b}$$

## B.4  Straight-Through

**Proposition 4.** Assume that pre-activations $a^k(x^{k-1}; \theta)$ are multilinear functions in the binary inputs $x^{k-1}$, and the objective $f$ is differentiable. Then, approximating $F_Z$ and $f$ linearly around their arguments for the sampled base state $x$ in Algorithm 1, we obtain the straight-through estimator in Algorithm 2.

*Proof.* Consider the last layer. The linear approximation to $f$ at $x^L$ allows to express

$$f(x_{\downarrow i}^L) \approx f(x^L) + \tfrac{\partial f(x^L)}{\partial x_i^L}(-2x_i^L); \tag{34}$$

$$df_i = f(x^L) - f(x_{\downarrow i}^L) \approx 2x_i^L \tfrac{\partial f(x^L)}{\partial x_i^L}. \tag{35}$$

We use the expression for the derivative of layer probabilities in parameters (9)

$$D_i^l(x) = p_Z(a_i^l)x_i^l \tfrac{\partial}{\partial \theta^l} a_i^l(x^{l-1}; \theta^l) \tag{36}$$

and its oddness in $x_i^l$. The gradient in parameters of the last layer becomes

$$\sum_i D_i^L df_i = \sum_j p_Z(a_i^L)\tfrac{\partial}{\partial \theta^L} a_i^L(x^{L-1}; \theta^L) 2 \tfrac{\partial f(x^L)}{\partial x_i^L}, \tag{37}$$

where we have canceled $x_i^L x_i^L = 1$.

With the linearization of $F_Z$ at $a^k$, the Jacobians $\Delta$ express linearly as follows:

$$\Delta_{i,j}^k = x_j^k\big(F_Z(a_j^k) - F_Z(a_{j\downarrow i}^k)\big) \approx x_j^k p_Z(a_j^k)\big(a_j^k - a_{j\downarrow i}^k\big), \tag{38}$$

where $p_Z$ is the noise density, *i.e.* derivative of $F_Z$, and we have denoted $a_{j\downarrow i}^k = a_j^k(x_{\downarrow i}^{k-1}; \theta^k)$. Because $a^k$ is linear in $x_i^{k-1}$, we have similarly to (34) that

$$a_j^k - a_{j\downarrow i}^k = 2x_i^{k-1} \tfrac{\partial}{\partial x_i^{k-1}} a_j^k(x^{k-1}). \tag{39}$$

This allows to express

$$\Delta_{i,j}^k = 2x_j^k x_i^{k-1} p_Z(a_j^k)\tfrac{\partial}{\partial x_i^{k-1}} a_j^k(x^{k-1}) = 2x_j^k x_i^{k-1} \tfrac{\partial}{\partial x_i^{k-1}} F_Z(a^k(x^{k-1})). \tag{40}$$

Note the occurrence of the formal derivative of $F_Z(a^k(x^{k-1}))$ in $x_i^{k-1}$, that will be the only derivative that is used in the ST algorithm.

Finally observe that for the derivative in parameters of layer $l$, estimated with the product $D^l \Delta^{l+1} \cdots \Delta^L df$ in PSA, for each $k \geqslant l$ the factors $x_{i_k}^k$ appear exactly twice and thus cancel. We recover the product of Jacobians without extra multipliers by $x_{i_k}^k$, which can be implemented with automatic differentiation as proposed in Algorithm 2. $\qquad\square$

The connection to STE pointed out by [31] can be seen as a special case of our construction for a single hidden layer, where local expectation gradients [30] apply and it suffices to linearize $f$.

## B.5  Complexity and Efficient Implementation of PSA

We have made the following complexity claim.

**Proposition 2.** The computation complexity of gradient by Algorithm 1 in all parameters of a network with convolutional and fully connected layers is the same as that of the standard back-propagation.

For fully connected networks this complexity is indeed linear in the total number of inputs and weights. For convolutional networks it is a bit more tricky because convolutions can generate big output tensors. In this case the complexity can be stated as linear in the total number of inputs, weights and hidden units.

We prove the claim by giving the algorithms how to implement all necessary computations with the same complexity as standard back-propagation.

First, we observe that the numbers $q$ in Algorithm 1 are only used to determine the derivative and set to zero value by line (8). The pre-activations $a$ in Algorithm 1 are of the same form as in the standard network, evaluating $F_Z$ component-wise does not increase complexity. Therefore backpropagation for these parts takes the same time. We only need to additionally compute the matrices $\Delta$ on the backward pass and explicitly implement the transposed multiplication (resp. transposed convolution) with them to define a custom backprop operation for the update (7).

**Fully Connected Layers**   Consider the case of a fully connected layer with pre-activation $a(x) = Wx + b$. Then $\Delta_{ij}$ has the same size as the matrix $W$. Recall it expresses as

$$\Delta_{i,j} = x_j^k \Big( F_Z(a_j) - F_Z(a_j(x_{\downarrow i}^{k-1})) \Big). \tag{41}$$

The first summand can be computed in linear time once $x^k$ and $a$ are known. The second summand is slightly more complex as it involves pre-activations for inputs with a flipped component $i$. For linear layers we have:

$$a_j(x_{\downarrow i}^{k-1}) = W x_{\downarrow i}^{k-1} + b = a_j - 2W_{j,i}x_i, \tag{42}$$

*i.e.* all the numbers $a_j(x_{\downarrow i}^{k-1})$ can be computed in time $O(n_k n_{k-1})$ for a matrix $W \in \mathbb{R}^{n_k \times n_{k-1}}$. It follows that computing matrix $\Delta$ takes $O(n_k n_{k-1})$ time, the same as the matrix-vector multiplication $W x^{k-1}$ for forward pass or the transposed multiplication for the backward pass.

**Convolutional Layer**   With a convolutional layer, the implementation is more tricky, because computing $\Delta$ in a matrix form is no longer efficient. Consider the convolution pre-activation

$$a_{o,j} = \sum_{c, @i} w_{o,c,i-j} x_{c,i}, \tag{43}$$

where $c$ and $o$ are input and output channels, $i$ and $j$ are 2d indices of spatial locations and $@i$ denotes that the range of $i$ is given by the output location $j$ and the weight kernel size: $j - h/2 \leqslant i \leqslant j + h/2$. This notation makes it more easier to match with the equations in the matrix form.

For the gradient in the standard network, a transposed convolution occurs:

$$\frac{\partial}{\partial x_{c,i}^{k-1}} = \sum_{o, @j} w_{o,c,j-i} \frac{\partial}{\partial x_{o,j}^k}. \tag{44}$$

For the gradient in PSA, we need to implement the following sum with $\Delta$:

$$\frac{\partial}{\partial q_{c,i}^{k-1}} = \sum_{o,j} \Delta_{o,c,j,i} \frac{\partial}{\partial q_{o,j}^k}, \tag{45}$$

where in the case of convolution and symmetric noise, $\Delta$ is given by

$$\Delta_{o,c,j,i} = x_{o,j}^k \Big( F_Z(a_{o,j}) - F_Z(a_{o,j}(x_{c,\downarrow i}^{k-1})) \Big). \tag{46}$$

Using that $a_{o,j}(x_{c,\downarrow i}^{k-1})$ itself is given by the convolution, we have

$$a_{o,j}(x_{c,\downarrow i}^{k-1}) = \begin{cases} a_{o,j} - 2w_{o,c,i-j} x_{c,i}^{k-1}, & \text{if } -h/2 \leqslant i \leqslant h/2; \\ a_{o,j}, & \text{otherwise.} \end{cases} \tag{47}$$

Therefore $\Delta_{o,c,j,i}$ has the same support in indices $i, j$ as the convolution, but it is different in that it cannot be represented as just a function of $i - j$. It can be interpreted as a convolution with a kernel, which is spatially varied, *i.e.* at all locations a different kernel is applied.

Due to the structure of $\Delta$, the sum (45) takes the same range of indices as the convolution, we may write:

$$\frac{\partial}{\partial q_{c,i}^{k-1}} = \sum_{o, @j} \Delta_{o,c,j,i} \frac{\partial}{\partial q_{o,j}^k}. \tag{48}$$

The elements of the kernel $\Delta_{o,c,j,i}$ are computed in $O(1)$ time, therefore the full backprop operation (45) has the same complexity as backprop with standard network (assuming small kernel size, where convolution uses straightforward implementation and not FFT).

We take one step further, to show that convolution with $\Delta$ for logistic noise needs literally the same amount of operations. Computing the convolution (45) with the first summand of $\Delta$ is easy. As it does not involve the index $i$, it reduces to multiplication in the output space (indices $o, j$), summation over $o$ and a convolution with just the support indicator in $j$.

It remains to compute the convolution (50) with the second summand of $\Delta$, *i.e.*:

$$\sum_{o,@j} x_{o,j}^k F_Z\big(a_{o,j} - 2w_{o,c,i-j}x_{c,i}^{k-1}\big)g_{o,j}, \tag{49}$$

where we denoted the gradient w.r.t. the output as $g$. The factor $x_{o,j}^k$ is easily accounted for, by introducing $\tilde{g}_{o,j} = g_{o,j}x_{o,j}^k$. We can further expand for logistic noise:

$$\sum_{o,@j} \frac{1}{1 + e^{a_{o,j}} e^{-2w_{o,c,i-j}x_{c,i}^{k-1}}} \tilde{g}_{o,j}. \tag{50}$$

Since $x_{c,i}^{k-1}$ takes only two possible values, for each input gradient coordinate $c, i$ we need the convolution with $e^{\pm 2w_{o,c,i-j}}$. We can precompute $A_{o,j} = \exp(a_{o,j})$, $W_{o,c,l}^{\pm} = \exp(\pm 2w_{o,c,i-j})$, *i.e.* the expensive $\exp$ operations need to be performed only for the output and the kernel alone, and not inside the convolution. For the dominant complexity part involving $c, o, i, @j$ indices, we only need to compute and aggregate

$$\sum_{o,@j} \frac{\tilde{g}_{o,j}}{1 + A_{o,j}W_{o,c,i-j}^{\pm}}. \tag{51}$$

Compared to standard convolution, this costs only one extra addition and division operation. We call the operation (51) a *ratio convolution* and implemented it in CUDA. Since our implementation is not fully optimized and we need to load twice as much data ($g$ and $A$) for the input and $W^{\pm}$ for the "kernel", the actual run-time is 3-4 times slower than that of cuDNN standard convolution.

**Head Function** For the head function $f(x^L)$ that is a composition of a linear layer $Wx^L + b$ and some fixed function $h$, in order to compute $f(x^L) - f(x_{\downarrow i}^L)$ we need again to form all pre-activations $a_j(x_{\downarrow i}^L)$ that takes $O(n_L K)$ time, where $K$ is the number of classes (more generally, the dimensionality of the network output). This is of the same size as the matrix $W$. Assuming that the final loss function $h\colon \mathbb{R}^K \to \mathbb{R}$ (*e.g.*, cross-entropy) takes time $O(K)$, the computation in the last layer has complexity $O(n_L K)$ as we need to call this function for all input flips. This is however still of the same complexity as size of the matrix $W$.

## C Details of Experiments and Additional Comparisons

In this section we describe details of the experimental setup and measuring techniques and offer some auxiliary experiments.

### C.1 Gradient Estimation Accuracy

We use the simple problem instance shown in Fig. B.1(a) with 2 classes and 100 training points per class in 2D and a classification network with 5-5-5 Bernoulli units. The data was generated as follows. Points of class 1 (resp. 2) are uniformly distributed above $y = 0$ (resp. below $y = \cos(x)$) for $x \in [-\pi/2, \pi/2]$. The implementation is available in gradeval/expclass.py. We have experimented with several configurations varying the number of units and layers. Generally, with a smaller number of units ARM gets more accurate and ST gets less accurate, but the overall picture stays. We therefore demonstrate the comparison on the 5-5-5 configuration.

The training progress is shown in Fig. C.2. In this problem unbiased estimators perform well and an extra variance reduction of PSA is not essential. To measure RMSE and cosine similarity errors in Fig. 2, we collect $T = 10^4$ total samples for each estimator. For each value of the number of samples $M$ shown on the x-axis, we calculate the mean and variance of an $M$-sample estimator by using the $T/M$ sample groups to estimate these statistics. The same $T$ samples are used to estimate the values for varying $M$. Towards $M \to T$ the estimates become more noisy, so the rightmost parts of the plots shouldn't be considered reliable.

Comparisons with additional unbiased methods is proposed in Fig. C.3 and Fig. C.4. We compare to the following techniques: 1. REINFORCE with the constant input-dependent baseline set to the true expected function value of the loss objective per data sample. This choice represents the best of what one can expect to get with input-dependent baselines constant or trained with a neural network as NVIL [18]. 2. MuProp [10], which uses a linear baseline around deterministically propagated points. The constant part of the baseline may also optionally set to the input-dependent true value. It is seen from the results that the linear baseline of MuProp does not improve accuracy in deep layers, possibly connected to the fact that the loss depends only on the state of the last layer and thus its linear approximation using states of the first layers is not helpful. It is also seen in Fig. C.4 that the linear baseline of MuProp degrades at 2000 epochs. For this experiment we used only $T = 2000$ samples.

Figure C.1: Additional plots to Fig. 4: validation losses and training accuracies.

## C.2 Deep Learning

**Dataset** The learning experiments are performed on CIFAR10 dataset[2]. The dataset contains a training set and a test set. Following the common approach, we withhold 5000 samples (10 percent) of the training set as a validation set. Since we do not perform any hyper-parameter tuning or model selection based on the validation set, it provides an independent and unbiased estimates.

**Augmentation** For Affine augmentation experiment we used random affine transforms that included shifts by $\pm 5\%$ and rotation by $\pm 10 \deg$, linearly interpolated. Flip&Crop is the commonly applied augmentation for this dataset. It consists of random horizontal flips and random shifts with zero padding by $\pm 4$ pixels (`transforms.RandomCrop(32, 4)`). Fro the training plots and loss values achieved we can hypothesize that this augmentation is more "diverse" and harder to fit that the Affine augmentation above.

**Network** For simplicity, we used a variant of All convolutional network (Springenberg et al. 2014), with strides instead of max pooling. The network structure is as follows:

```
ksize = [3, 3, 3, 3, 3, 3, 3,  1 ]
stride= [1, 1, 2, 1, 1, 2, 1,  1 ]
depth = [96, 96, 96, 192, 192, 192, 192, 192]
```

There is no padding and the output is a $192 \times 2 \times 2$ binary tensor, which is then flattened and passed to the head function consisting of an affine transform to the 10-dimensional class logits. This network is smaller than VGG-type networks commonly used [12], however significantly smaller is size esp. considering the last fully connected layer. Our FC has size $768 \times 10$, the ones in [12] (there are three) are: $8192 \times 1024$, $1024 \times 1024$, $1024 \times 10$. No batch normalization is used for the purity of comparison experiments. If we were chasing the highest accuracy, we can confirm that BN improves the results.

**Optimizer** For the optimization we used batch size 64 and SGD optimizer with Nesterov Momentum 0.9 (pytorch default) and a constant learning rate. Because different methods have different variance and biases, for a fair comparison we tried to find the optimal learning rate for each method individually. We selected the learning rate by a numerical optimization based on the performance of the model in 5 epochs as measured by the objective optimized by the method (*i.e.* the sample-based estimate of the expected loss or the approximated expected loss) on the training set. We used exponentially weighted average on the objective value to reduce its variance. We used

```
scipy.optimize.minimize_scalar(f, method='bounded', bounds=(-6,
    0),maxiter=10)
```

for the numerical search of the optimal $\log$ of the learning rate. Arguably, this learning rate selection optimizing the short-horizon performance may be sub-optimal in a longer run, but is the first best approximation to deal with this issue.

Parameters of linear and convolutional layers were initialized as uniformly distributed. We then perform one iteration, computing mean and variance statistics over a batch and spatial dimensions and whitening pre-activations using these statistics, similar to batch normalization. This is performed only as a data-dependent initialization to make sure that activations are in a reasonable range on average and the gradients are initially non-vanishing.

**Test Metrics**   In our experiments all hyperparameters including the learning rates are tuned exclusively on the training set as detailed above. Hence the validation set provides an unbiased estimate of the test error and we report only it.

**Methods**   Here we specify additional details on the baseline methods used in the learning experiment. The ADF method, called AP2 in [**?** ], propagates means and variances through the hidden layers fully analytically. This method is the equivalent of the PBNET method in [22] when the weights are deterministic. We only sample the states of the last binary layer are sampled, as a general solution suitable with different head functions. The ADF family of methods includes expectation propagation [17] designed for approximate variational inference in graphical models. It computes an approximation to summation (5) by fitting and propagating a fully factorized approximation to marginal distributions $p(x^k|x^0)$ with forward KL divergence. The gradient of this approximation is then evaluated. The PSA method differs in that it approximates the gradient directly and does not make a strong factorization assumption. From the experiments we observe that ADF performs very well in the beginning, when all weights are initially random and then it over-fits to the relaxed objective.

In the `LocalReparam` method we sample pre-activations from their approximated Normal distribution and computed probabilities of the outputs analytically. This is related but different from the PBNET-S method [22], which samples activations from the concrete relaxation distribution and is not directly applicable to real-valued deterministic weights. The implementation of our methods and these baselines is available.

**Infrastructure**   The experiments were run on linux servers with NVIDIA GTX1080 cards and Tesla P100 cards.

**Running Time**   We measure running time for a single batch of size 64 and the running time for one epoch of training / validation. The training loop includes measuring statistics using 10 samples of the model per data point, which increases amount of forward computations performed. Please consider that these numbers are indicative only, as we did not optimize for performance beyond providing the CUDA kernel for PSA. We also have lots of overhead from implementing padding and strides in pytorch, externally to the kernel. Times are given in seconds.

| Method | Forward batch | Backward batch | Train with measurements | Epoch | Val Epoch with measurements |
|---|---|---|---|---|---|
| PSA | 0.013 | 0.056 | 105 | | 14 |
| Gumbel | 0.009 | 0.012 | 77 | | 13 |
| Tanh | 0.005 | 0.011 | 77 | | 12.5 |
| REINFORCE | 0.014 | 0.009 | 77 | | 14 |
| ADF | 0.012 | 0.031 | 91.5 | | 14 |
| LocalReparam | 0.015 | 0.029 | 98 | | 14 |
| ST | 0.009 | 0.011 | 77 | | 13 |
| HardST | 0.009 | 0.011 | 77 | | 13 |

### C.3   MuProp and REINFORCE with Baselines

For fairness of comparison we have additionally tried to use REINFORCE with a centering variance reduction and MuProp. As a baselines for these methods we use the exponentially weighted average (EWA) of the loss function per data point with momentum=0.9. Since the learning rate needed for these methods is rather small, the EWA is close to the true expected loss value per data point. Since the EWA is kept per data point, this is an accurate input-dependent constant baseline. We made several learning trials with automatically selected learning rates (denoted with stars in Fig. C.7) as well as manually set learning rates. Since automatic learning rates appear to be too high as they lead do divergence, it only makes sense to decrease these learning rates. We tried setting the learning rates smaller. The results in Fig. C.7 show that with some careful choice of parameters, REINFORCE and MuProp can make progress. However, the learning rates required are 1-2 orders smaller than with biased methods, which leads to much slower performance. Indeed, comparing Fig. C.7 and Fig. 4 (note the

Figure C.2: Training losses when training a small network 5-5-5 on the data in Fig. B.1(a) with several methods. Here the dashed curves show the SBN objective, the expected loss of randomized predictor (15), which upper bounds the the loss of the ensemble (15) shown with solid curves. We see that REINFORCE (score) does not do very well in the beginning, but eventually gets into a mode that allows it to optimize the objective. The methods ARM and PSA perform similar and ST is somewhat slower due to its bias for this small model.

Figure C.3: Root mean squared error of the gradient in layers 1 to 3 relative to the true gradient length after epoch 1. Comparisons with more unbiased methods in the same setting as Fig. 2.

Figure C.4: Cosine similarity of the estimated gradient to the true gradient in layer 1 at different points during training. Comparisons with more unbiased methods in the same setting as Fig. 3.

scale difference of $y$-axis), we observe that variance-reduced REINFORCE and MuProp in 1000 epochs have barely made the progress of biased estimators in 10 epochs.

Figure C.5: Dependence on the network width. Shown gradient RMES in Layer 1 after epoch 1 (close to randomly initialized network). Bias of PSA and ST is more prominent in small models.

Figure C.6: Dependence on the network depth. Shown gradient RMES in Layer 1 after epoch 1 (close to randomly initialized network). The variance of unbiased estimators grows much faster when increasing the network depth. The network width is fixed to 5 units per layer.

Figure C.7: Performance of REINFORCE and MuProp with different learning rates and input-dependnet constant baselines estimated using a running averages. No smoothing is applied across epochs in this plot. Note the limits of $y$-axis in comparison to Fig. 3.

## Footnotes

[2]`https://www.cs.toronto.edu/~kriz/cifar.html`