[Reviews · NeurIPS 2020]

Review 1

Summary and Contributions: This work proposes a new method (PSA) that applies local expectation (a.k.a. Rao-Blackwellization) to gradient estimation for deep binary discrete latent variable models. The challenge here is the dependency across layers. Previous works rely on local importance weighting (Titsias & Lázaro-Gredilla, 2015) or Gumbel-max reparameterization (Tokui & Sato, 2017) to address the challenge. The contribution of this work is to instead propose a linear approximation to simplify the local expectation estimator. This method (PSA) is biased but allows to compute the gradients in parameters of all layers in a single backward pass. A further linear approximation step of the PSA leads to a deep straight-through estimator. Experiments on deep stochastic binary networks show that the proposed method outperforms strong competitors when affordable number of samples are used.

Strengths: The rebuttal resolves my concern on the comparison with Tokui & Sato (2017). I raised the rating accordingly. ---------------- original review below ---------------- * This paper addresses an important open challenge in training hierarchical discrete (binary here) latent variable models and propose a fast, easy-to-implement gradient estimator based on a linear approximation that resolves dependencies across layers. * The empirical evaluation is sound, with a number of baselines (REINFORCE, ARM, Concrete) considered. And the proposed method consistently outperforms baselines in training. Figure 2 is particularly convincing, as shown by the difference of biased/unbiased estimators, and PSA has no bias in layer 3, which behaves similarly as unbiased methods. * The background material on gradient estimators is well-organized, including very closely related work such as Titsias & Lázaro-Gredilla (2015), Tokui & Sato (2017).

Weaknesses: The paper is technically sound and I have no major comments on the weaknesses. Some minor points & discussion: * L109: ``We derive a general form of STE under a very clear approximation extending the linearization construction [31] '' The authors tend to summarize the contributions of this paper as extending the approach of Tokui & Sato (2017). However, a notable difference is that Tokui & Sato apply Gumbel-max reparameterization to all discrete latents to resolve the dependencies, which is not the approach taken here. I believe using Tokui & Sato's method we can also train deep binary stochastic networks with a manageable cost (linear to the number of bernoulli latents). Therefore, it is important to discuss their differences and advantages of the linear approximation used in this work (it also adds bias). Experiments on comparing them are very welcome, too. * L100: Finally, many experimentally oriented works successfully apply straight-through estimators (STE), originally considered by Hinton for variational auto-encoders. The statement here is imprecise, it is proposed for deep autoencoders with binary hidden codes. No variational bounds were used.

Correctness: Yes.

Clarity: The paper is well-written.

Relation to Prior Work: I'm happy to see that Tokui and Sato (2017) is cited in the related work, which is a less-known, under-appreciated work in gradient estimation literature. Having put that, I'd like to see more discussion on the comparison between their method and the method proposed here. I can see that their method to resolve dependencies between latent variables are different, by utilizing the Gumbel max reparameterization.

Reproducibility: Yes

Additional Feedback: L47: Figure 1 bottom should be Figure 1 right? The same in L70.


Review 2

Summary and Contributions: In this paper, the author proposes a method to estimate the gradients in the stochastic binary networks. In the simulation study, the author shows their method has more accurate gradient estimation, under the measurement of MSE.

Strengths: The paper studies the problem of gradient back-propagation through stochastic binary layers, which is an important direction for model compression and interpretable deep learning.

Weaknesses: 1. The experiments setups are not persuasive. For the gradient estimation accuracy, the author conduct experiment only on 2 classes 2D simulation data. The author does not mention how the 100 training data generated, which is in quite a small amount even in the simulation study. The network is in special design as 5-3-3 Bernoulli cases, which is insufficient to conclude the proposed method is better in gradient estimation. The reviewer expects to see more simulation results by varying the unit number in each layer. 2. The performance on the real-world dataset is not satisfying enough. The PSA method seems not to achieve the best accuracy or the fastest convergence. The ST method is previously proposed, which I think cannot be recognized as the author's contribution. Besides, only the validation results are reported. What is the performance on the testing set? 3. Important baselines are not compared. The ARM gradient is a competitive baseline, which the author only compared under a special simulation setup. What is the reason that the author does not compare with ARM on the CIFAR classification task? 4. The proposed PSA method requires more computation than baselines. In algorithm 1, when feeding forward, the PSA requires the calculation of all the flipped previous layer output into the current layer. The comparison of computation complexity is expected in the experiment part. 5. The notations of the method are difficult to follow. It is much better if the authors can begin their analysis with 1-hidden-layer SBN first, which will simplify the notation a lot.

Correctness: The empirical methodology is not convincing enough to the reviewer. More experiment details are needed to support the author's conclusion.

Clarity: The main idea of the proposed method is easy to follow. The experimental part needs improvement to be well-organized.

Relation to Prior Work: The Path Sample-Analytic is novel from previous work. But the variant ST method already previously appeared.

Reproducibility: No

Additional Feedback: Thanks to the author's response. I have upgraded my score. However, I strongly suggest the author conduct more experiments to make a solid submission.


Review 3

Summary and Contributions: EDIT: I have read the authors' feedback and maintain my original assessment. --- This paper presents a new estimation method for the gradient of the expected model response of Stochastic Binary Networks (SBN) w.r.t. the parameters. The gradient estimator is biased but significantly reduces variance, as evidenced through a careful analysis of what terms in the gradient are estimated or approximated by linearization, as well as careful empiricism in both small and moderate scale experiments. The method stabilizes training.

Strengths: A key contribution of this paper is its exceptionally clear exposition of the SBN model as well as a careful analysis of their novel estimator, which they call a "path-sample analytic" (PSA) method due to its explicit summation over many different dependency paths in a Bayesian-network representation of an SBN. Using their analysis, the authors also give a rigorous justification of the straight-through estimator as a particular linearization and noise-model assumption that yields exactly the form of the straight-through estimator in the backwards pass. This knowledge could be used to swap in different assumptions and different approximations to yield other biased but potentially lower-variance methodologies. It also yields a pleasing explanation of the straight-through estimator on the basis of comprehensible mathematical approximations and simplifying assumptions. This work is also carefully positioned with respect to prior gradient estimators, including various relaxation and control-variate-based methodologies that attempt to correct the high variance of the REINFORCE estimator. A final contribution is the idea behind the PSA algorithm itself, which may well inspire many new methods for reducing the variance of various other Bayesian networks with discrete nodes. Although it does simply reduce to a clever application of Rao-Blackwellization, e.g., a local expectation gradient, that exploits the small number of states of a given binary random variable, this approach of paying a small (but mathematically comprehensible) bias cost for a large reduction in variance is novel and useful. I found the presented empiricism to be very good. The careful analysis of the proposed estimators versus true gradient was very informative, and allowed the authors to discuss limitations of the method.

Weaknesses: While the empiricism that was done was quite good, I am surprised at the choice only of CIFAR-10, when this method appears to allow gradient estimation through much deeper networks without losing coherence of the gradient signal due to noise. I would have appreciated more ambitious experiments beyond simply comparing to other baselines.

Correctness: The claims and methods appear correct. The empirical methodology establishes the advantages and disadvantages of this method over other true gradient estimators and relaxations. At one point, the authors refer to Hinton's discussion of variational autoencoders, which is wrong. I am familiar with the lecture in question and the model was simply a deep autoencoder (variational autoencoders were invented some years later).

Clarity: The paper is well written and exceptionally clear in its mathematical exposition, with a few scattered typos: 41 reminder <-> remainder 78 we <-> We 246 With <-> We 280 It's <-> its There are likely more that I did not notice while reading.

Relation to Prior Work: Yes, the work clearly discusses relation to previous contributions.

Reproducibility: Yes

Additional Feedback: Discussion of extensions beyond logistic noise and with other approximations in the gradient would be of value. Similarly, an example where the linearization approximation induces a bias that makes convergence impossible. These seem like straightforward corollaries to this work. Discussion also of what other interesting Bayesian networks the PSA methodology might apply to would be of value.

[Author Response · NeurIPS 2020]

**R2: Relation to Tokui & Sato (2017) [31]**. Their RAM estimator for each discrete variable and each its state needs to recompute the states of all dependent variables [31, Alg.1] and therefore scales quadratically with the number of variables despite the sample of the noises being drawn only once. It is proposed in order to study the quality of control variate techniques. We have compared with several tractable control-variate techniques in Appendix Fig. C.3 and Fig. C.5.

The Gumble-max reparametrization resolves dependencies between latent noises only. When flipping the discrete state $z_i$ for fixed latent noises $\epsilon_{-i}$, the dependent discrete variables and the objective function may still change causing the quadratic complexity of [31 Alg.1]. To avoid confusion, we should clarify that the Gumble-max reparametrization is not differentiable and the "reparameterization trick" is never used in their RAM method in contrast to what the abstract says.

In context of L109 "extending the linearization construction [31]" we mean extending their derivation of Straight-Through [31 sec. 6.4], where it is assumed that the loss function $f$ is differentiable in each discrete variable $z_i$ (and does not depend on it through a chain of other discrete variables) and thus can be linearized [31 eq. 8]. This derivation is applicable to one layer only, in which case it matches our ST. Hence there is nothing to compare experimentally. This result is a side observation in this paper, indeed not noticed in alternative explanations of ST [4, 36].

**R2, R4: Variational autoencoders**. Thanks, we will correct to "deep autoencoders with stochastic binary codes".

**R3: Experimental setup.** We will describe the generation procedure. Points of class 1 (resp. 2) are uniformly distributed above $y = 0$ (resp. below $y = \cos(x)$). The implementation is available in gradeval/expclass.py. The data is shown in Fig. B.1 (a). We have experimented with several configurations varying the number of units and layers. Generally, with a smaller number of units ARM gets more accurate and ST gets less accurate, but the overall picture stays. The displayed results are actually for a 5-5-5 configuration as described in Appendix C.1 (mentioning 5-3-3 in L224 is a typo). We will extend the appendix to show more cases varying the number of units / layers.

**Performace.** We address only the training performance and not the generalization performance, which in practice involves batch norm, pretraining and architecture search. However the method does achieve the best accuracy and the fastest convergence in iterations in comparison with other training methods under the same setup.

**The ST method is previously proposed**. We disagree. As we discuss in the related work, it has been proposed as a practical hack in several different variants. Other works attempted studying its properties. We derive it for deep models. In our view we are the first to propose it as a formal method.

**Test set performance.** In our experiments all hyperparameters including the learning rates are tuned exclusively on the training set (see Appendix C.2). Hence the validation set provides an unbiased estimate of the test error.

**ARM on CIFAR.** ARM has a prohibitive complexity for deep models (see L90). We expect it to have high variance for deep models as well. See also appendix L643-655 comparing to MuProp.

**Computation complexity.** Our complexity analysis (Proposition 2, proof in Appendix B.5) shows that the required computation for all flips has the same complexity as standard forward propagation. Additionally, in Appendix B.5 we show how to overload backprop operations to achieve the FLOPs complexity as low as 2x standard backprop. Additionally, in L664 of appendix we report all running times with the currently provided implementation (using GPU but suboptimal).

**Single layer case.** The single layer case is well covered in the literature [5, 30, 31], we also detail it in Appendix B2 (L453-461).

**R4. More advanced experiments.** We face here the situation that ST methods have been already applied successfully to deep residual networks, e.g. on ImageNet. We do not expect to beat them simply by an improved gradient estimator without dealing with learning schedules, pretraining, designing special architectures, etc. We will explore this and other applications in the future work.

**Beyond logistic noise.** All theory applies seamlessly to any continuous noise distribution. The logistic noise is really being used only in eq. (49-51) to optimize the implementation (affects constant complexity factors).

**Feedback.** Thanks, we will discuss limitations and possible applications.

[Meta-Review · NeurIPS 2020]

The paper proposes a new gradient estimator (PSA) for stochastic binary networks, based on combining the local expectation gradients (LEG) idea with a linear approximation. PSA scales linearly in the network size, and thus is much faster than LEG. While the linearization step means that the PSA gradient is biased, the empirical analysis on a small network shows that for almost any reasonable computational budget PSA has the lowest the overall RMSE in the gradient estimate out of several biased and unbiased estimators. The paper is well written and the proposed method is clearly described. The algorithm is simple and might become a popular alternative to the less accurate straight-through estimator. Overall, a solid contribution to the gradient estimation literature.